# Risks in Induction of Platelet Aggregation and Enhanced Blood Clot Formation in Platelet Lysate Therapy: A Pilot Study

**DOI:** 10.3390/jcm11143972

**Published:** 2022-07-08

**Authors:** Ying-Hao Wen, Chen-Fang Lee, Yu-Ju Chen, Gwo-Jyh Chang, Kowit-Yu Chong

**Affiliations:** 1Department of Laboratory Medicine, Linkou Chang Gung Memorial Hospital, Taoyuan 33305, Taiwan; b9209011@cgmh.org.tw; 2School of Medicine, National Tsing Hua University, Hsinchu 30013, Taiwan; 3Graduate Institute of Clinical Medical Sciences, College of Medicine, Chang Gung University, Taoyuan 33302, Taiwan; 4Department of Liver and Transplantation Surgery, Linkou Chang Gung Memorial Hospital, Taoyuan 33305, Taiwan; chenfanglee@gmail.com; 5Department of Neurology, Mackay Memorial Hospital, Taipei 10449, Taiwan; yuju_510@hotmail.com; 6Department of Medical Biotechnology and Laboratory Science, Chang Gung University, Taoyuan 33302, Taiwan; 7Graduate Institute of Biomedical Sciences, College of Medicine, Chang Gung University, Taoyuan 33302, Taiwan; 8Hyperbaric Oxygen Medical Research Lab, Bone and Joint Research Center, Chang Gung Memorial Hospital, Linkou, Taoyuan 33302, Taiwan

**Keywords:** platelet concentrates, platelet-rich plasma, thrombosis, blood clot, rotational thromboelastometry

## Abstract

Platelet concentrates (PCs) are widely used in regenerative medicine; as it is produced from freeze–thawing PC, platelet lysate (PL) has a longer shelf life. The thrombotic risk of PL therapy needs to be explored since PL and PC contain cytokines that contribute to platelet aggregation and thrombus formation. Whole blood samples of 20 healthy subjects were collected; PL was produced from PCs with expired shelf life through freeze–thawing. The direct mixing of PL with platelet-rich plasma (PRP) or whole blood was performed. In addition, rotational thromboelastometry (ROTEM) was used to investigate whether PL enhanced coagulation in vitro; the effects of fibrinogen depletion and anticoagulants were evaluated to prevent hypercoagulation. The results showed that PL induced platelet aggregation in both PRP and whole blood. In ROTEM assays, PL was shown to cause a significantly lower clotting onset time (COT) and clot formation time (CFT), and a significantly greater α angle and maximum clot firmness (MCF). Compared with the controls, which were 1:1 mixtures of normal saline and whole blood, fibrinogen depletion of PL showed no significant difference in CFT, α angle and MCF. Moreover, heparin- and rivaroxaban-added PL groups demonstrated no clot formation in ROTEM assays. Platelet lysate-induced hypercoagulability was demonstrated in vitro in the present study, which could be prevented by fibrinogen depletion or the addition of an anticoagulant.

## 1. Introduction

Platelet concentrates (PCs) are classified as platelet-rich plasma (PRP), platelet-rich fibrin (PRF) or concentrated growth factors according to their preparation methods and fibrin composition [1,2]. PCs are widely used in the treatment of osteoarthritis, tendinopathy, chronic wound and bone injuries due to their richness in growth factors and cytokines with regeneration potential [3,4,5]. Moreover, meta-analyses have demonstrated that PCs improve functional status and relieve pain in patients with osteoarthritis or tendinopathy [4,5].

The composition of growth factors and cytokines differs in different types of PCs [6], and >300 proteins are released by activated platelets [7]. Therefore, the standardization and personalization of PCs in the treatment of specific disease is necessary. The storage condition of PCs is also an important issue when multiple injections of PCs are required [8]. In a blood bank, PCs need to be stored in 22 °C incubators with agitation, rendering a shelf life of 7 days [9]. Levels of vascular endothelial growth factor (VEGF), fibroblast growth factor (FGF)-basic, hepatocyte growth factor, insulin-like growth factor-1, platelet-derived growth factor (PDGF)-AB, endothelial growth factor (EGF) and transforming growth factor (TGF)-beta1 are sustained in PCs for 7 days [10]; however, their short shelf life does not meet clinical needs.

Platelet lysate (PL) is produced from freeze–thawing PC, and growth factors and cytokines are completely released from the platelets without additional activating agents [11,12,13]. The standardization of freeze–thawing procedures is feasible, and low batch-to-batch variability in the concentration of growth factors or cytokines is accomplished under good manufacturing practice (GMP)-compliant manufacturing [14]. Moreover, concentrations of PDGF-BB and TGF-beta are stable in PL under −20 °C storage for 1 year [11], and the levels of VEGF, EGF, and FGF-basic are sustained under −80 °C storage for 2 years. Therefore, multiple injections of stored PL are applicable in a single preparation.

PL and PC contain cytokines involved in platelet aggregation and thrombus formation, and arterial thrombotic occlusion causes ischemia or infarction [15]. The implementation of PL and PC in the treatment of diseases of internal organs is under development or being investigated in pilot studies [16,17,18,19,20,21,22,23,24,25]. On the other hand, irreversible blindness caused by ophthalmic artery occlusion was found in several patients following PC injection into facial skin [26,27]. This observation has led us to draw a parallel and make the assumption that PL could possibly cause thrombosis. Furthermore, since the literature shows that direct injections of PC into target organs are the main route of application [16,17,18,19,20,21,22,23], risks of thrombosis arising from PL therapy need to be carefully assessed.

To investigate whether PL causes thrombosis, we divided the study into three parts—platelet aggregation, blood clot formation, and prevention of hypercoagulation. The effects of PL on platelet aggregation through direct mixing of PL with PRP and PL with whole blood were evaluated in vitro to further explore the potential hypercoagulability induced by PL. In addition, whether PL enhances blood clot formation via rotational thromboelastometry (ROTEM) was investigated. Subsequently, the use of fibrinogen depletion and anticoagulants to prevent hypercoagulation was also assessed.

## 2. Materials and Methods

### 2.1. Study Subjects and Platelet Lysates

Twenty healthy volunteer donors were included in this study. From each donor, 20 mL of whole blood samples was collected in a sodium citrate vacuum tube. Ten samples were used in platelet aggregation assays, and the others in ROTEM assays. PL was produced from PCs with an expired shelf life obtained from the Taiwan Blood Services Foundation, which is a non-remunerated blood donation system. The basic quality of platelet concentrations >2.75 × 10^10^ per bag (250 mL) of the PC supplies was assured by a hematological analyzer XN-9000 (Sysmex Corp., Kobe, Japan) (data not shown). On the other hand, PLs induce platelet aggregation too quickly to be measured by light transmission aggregometry. Nonetheless, a decrease in platelet concentration was consistently found in PLs throughout the test to support the thrombotic potential of PLs. PCs were frosted at −80 °C overnight and thawed at 37 °C. After centrifugation at 3000 *g* for 10 min, the supernatant, now designated as PL, was stored at −80 °C. Commercial fibrinogen-depleted human PLs (HELIOS UltraGRO™-Advanced from AventaCell BioMedical Corp., NW Atlanta, GA, USA and NEBHPL-100 from Nautia Evo Burgeon, Taipei, Taiwan), were used for ROTEM assays.

### 2.2. Platelet Aggregation Assay

Effects of PL on platelet aggregation were investigated in PRP and whole blood. PRP was prepared by centrifuging whole blood at 190 *g* for 10 min. PL was mixed with PRP to achieve the safe and pretested minimal effective PL mixture concentrations of 2.5%, 5% and 10% (data not shown). Normal saline was mixed with PRP, in a 1:10 mixture, for use as a negative control, and 2.0 mg/mL ristocetin was added in PRP as a positive control. After 15 min, changes in platelet concentration in different PL concentrations and large cell ratio of platelet were evaluated by a hematological analyzer XN-2000 (Sysmex Corp., Hyogo, Japan). The number of platelets per aggregate was obtained by microscopic readings to compare the effect of PL in different concentrations. Percentage change in platelet concentration in PRP was defined as:

([platelet concentration of PL mixture measured by hematological analyzer-calculated platelet concentration of PL mixture]/calculated platelet concentration of PL mixture) × 100%.

PL was also added to the whole blood, and concentrations of the resulting PL mixtures were 30%, 40% and 50%, which were determined in pretests (data not shown). The negative and positive controls used and data evaluations were performed as described in PRP above.

### 2.3. Rotational Thromboelastometry Assay

ROTEM assay was performed to investigate whether PL enhanced coagulation. In addition, commercial fibrinogen-depleted human PL and anticoagulant-added PL HELIOS UltraGRO™-Advanced and NEBHPL-100 were used in ROTEM assays to assess fibrinogen depletion and anticoagulants-induced prevention of hypercoagulation. Before performing the INTEM test of ROTEM assay in ROTEM^®^ delta (Werfen; Barcelona, Spain), PL was added to whole blood at 50% PL concentration of the mixture, as were the commercial fibrinogen-depleted human PL and anticoagulant-added PL. Normal saline was mixed with whole blood in a 1:1 ratio for use as a control. Clotting onset time (COT), clot formation time (CFT), α angle and maximum clot firmness (MCF) were measured according to manufacturer’s instructions. Anticoagulant-added PL was prepared by mixing PL in heparin tubes (BD, Franklin Lakes, NJ, USA) and by mixing 15 mg ground rivaroxaban (Xarelto) with 1 mL PL.

### 2.4. Statistical Analysis

Differences in changes in platelet concentration, large cell ratio of platelet, and the number of platelets per aggregate between different PL concentrations in the platelet aggregation assay, were analyzed for significance (*p* < 0.05) by Student’s *t*-test. COT, CFT, alpha angle and MCF in ROTEM assay were also analyzed for significance (*p* < 0.001) by Student’s *t*-test. Statistical analysis was performed using SPSS (version 20; SPSS Inc., Chicago, IL, USA).

## 3. Results

### 3.1. Platelet Lysate Enhances Platelet Aggregation in Platelet-Rich Plasma

Important changes in platelet parameters were investigated in mixtures of platelet lysate and platelet-rich plasma or whole blood. To investigate whether PL enhances platelet aggregation, the direct mixing of PL with PRP was performed. Platelet concentration was significantly decreased from 16.3% to 39.8% in the tested 2.5–10% PL and PRP mixtures when platelet lysate was mixed with platelet-rich plasma in increasing concentrations (Figure 1A). The platelet concentration percentage change increased as the PL concentration rose. The large cell ratio of the platelet was significantly increased from 25.0% to 32.1% in the 5% and 10% PL and PRP mixture (Figure 1B). Under the microscope, platelet aggregation was observed in the PL-PRP mixture (Figure 1C), as ristocetin also induced platelet aggregation in PRP in the positive control samples (Figure 1D). Moreover, the number of platelets per aggregate also significantly increased by 79.2% in the PL-PRP mixture (Figure 1E). The findings indicate that PL enhanced platelet aggregation in PRP. For comparative purposes, the data in Figure 1 are summarized in Table 1.

### 3.2. Platelet Lysate Enhances Platelet Aggregation in Whole Blood

To further explore whether PL enhances platelet aggregation, direct mixing of PL with whole blood was performed. When platelet lysate was mixed with whole blood, 40% and 50% PL significantly decreased the platelet concentration (Figure 2A). The large cell ratio of platelet was significantly increased in 30%, 40%, and 50% PL and whole blood mixture (Figure 2B). Platelet aggregation was also observed in PL-whole blood mixture and ristocetin-added whole blood under the microscope (Figure 2C,D). The number of platelets per aggregate was also significantly increased in PL-whole blood mixture (Figure 2E), indicating the PL enhancement of platelet aggregation in whole blood. The datum in Figure 2 are summarized in Table 2.

### 3.3. Platelet Lysate Enhances Blood Clot Formation

To assess potential hypercoagulation induced by platelet lysate, the ROTEM assay was used. The results show that 50% PL induced a significantly earlier onset of blood clot formation than the control, which was a 1:1 mixture of normal saline and whole blood (Figure 3A); 50% PL also induced a significantly rapid 20 mL blood clot formation as indicated in the shortened clot formation time (Figure 3B). More rapid clotting in the 50% PL samples was also shown by significantly larger α angles than those found in the control (Figure 3C); significantly greater MCF induced by 50% PL was also noted (Figure 3D). These results demonstrated that PL induced hypercoagulation.

### 3.4. Platelet Lysate-Induced Hypercoagulability Is Prevented by Fibrinogen Depletion or Addition of an Anticoagulant

To prevent hypercoagulation induced by platelet lysate, fibrinogen depletion of PL and the effects of an addition of anticoagulant to PL were evaluated in ROTEM assays using two commercial sources of platelet lysates. Significant earlier blood clot formation was induced by the commercial fibrinogen-depleted PL from HELIOS UltraGRO™-Advanced (Figure 3A). CFT, α angle, and MCF showed no significant difference in the fibrinogen depletion of PL groups compared with that of the control (Figure 3B,C). Moreover, heparin-added and rivaroxaban-added PL groups demonstrated no clot formation (data not shown). The results show that the fibrinogen depletion of PL and addition of heparin or rivaroxaban could minimize hypercoagulation induced by PL.

## 4. Discussion

Dense granules and α-granules of platelets contain small molecules, such as adenosine di- and triphosphate, von Willebrand factor, fibrinogen and coagulation factors, which participate in primary (platelet aggregation and activation) and secondary hemostasis (coagulation) [28,29]. Evidence of platelet lysates being rich in these hemostasis molecules leading to potential PL-induced thrombosis was revealed in the present study. We showed that PL induced platelet aggregation in the use of platelet-rich plasma or whole blood (Figure 1 and Figure 2; see also summary in Table 1 and Table 2), and PL-enhanced coagulation in ROTEM assays (Figure 3).

Among patients complicated with visual loss after facial PRP injection, acute occlusion of ophthalmic artery or central retinal artery was found; multiple subacute brain infarcts were also noted [26,27]. Platelet activation and degranulation are induced by vascular injury when PRP is injected into target tissues by a needle [28,29]. Platelet aggregation and thrombus formation could occur in the vicinity of the PRP injection site, and detached blood clots could block blood vessels and cause tissue infarction. The collective results of the present study are consistent with these previous reports and should raise an awareness of thrombotic risks arising from clinical use of PL and PC.

To prevent hypercoagulation caused by PL, the use of fibrinogen-depleted or anticoagulant-added PL may be safer in practical applications, as supported by data presented in the present study (Figure 3). Fibrinogen-depleted PL showed significant reduction in clot formation time, α angle and maximum clot firmness compared with data obtained using PL, and heparin- and rivaroxaban-added PL, which demonstrated no clot formation in ROTEM assays. Fibrinogen-depleted PLs are commercially available; however, the preparation of autologous fibrinogen-depleted PL for personalized therapy is needed [30]. Sterile heparin-added PL could be conveniently prepared by mixing PL in heparin collection tubes, with heparin dosages far less than those used in antithrombotic and thrombolytic therapy [31].

The kinetics and strength of clot formation are provided by ROTEM assay and are used to guide blood transfusion in patients who undergo cardiac surgery, liver transplantation, and other bleeding situations [32]. In this study, hypercoagulation was detected in the whole blood samples with the addition of 50% PL, mimicking the hypercoagulation status caused by PL injected into the blood vessels (Figure 3). Therefore, COT, CFT, α angle, and MCF of ROTEM assay could be risk indicators for PL therapy.

In the field of regenerative medicine and cell therapy, PL-based biomaterials are developed for tissue regeneration and wound healing [33,34]. Moreover, human PL has a similar potential in promoting cell growth and expansion as fetal bovine serum. Thus, human PL is considered as an alternative cell culture supplement for cell therapy [35,36,37,38]. Human PL also has the advantages in the xenogenic-free manufacturing of cellular therapy products after the consideration of issues with infectious safety, potency and functionality [35,36,37,38]. The results of the present study predict that a hypercoagulable state would be imposed on patients receiving intravenous administrations of cellular products with PL supplements. Animal studies are needed to further substantiate our proposition. Furthermore, we should also be wary of variations in PL concentrations resulting from different sources and preparation protocols of PC being used; such variations would impose different risk levels in thrombosis in clinical PL injections.

## 5. Conclusions

Platelet concentrate and lysates have been widely applied in regeneration medicine, and PL has the advantages of being stable in long-term storage and GMP compliance achievement. Hypercoagulability induced by PL was revealed in the present study, which could be prevented by fibrinogen depletion or anticoagulant addition. If further parameters of the ROTEM assay could be used to evaluate and monitor the hypercoagulable state, then PL applications could be further developed for safer use in regenerative medicine.

## Figures and Tables

**Figure 1 jcm-11-03972-f001:**
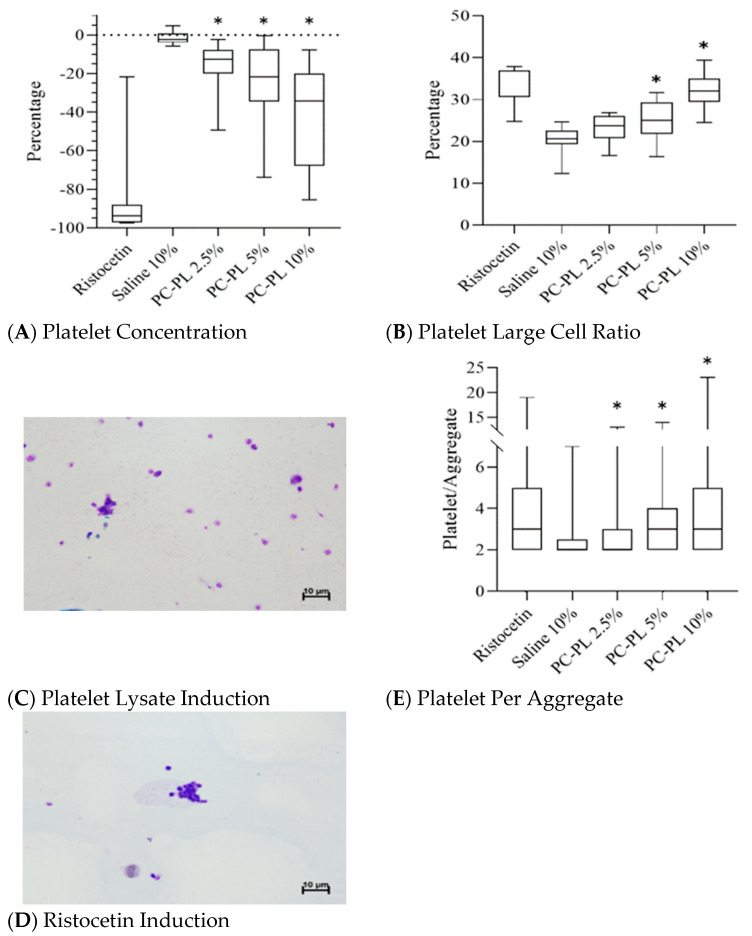
Effects of platelet lysate on platelet aggregation in direct mixing of platelet lysate with platelet-rich plasma. Percentage changes in platelet concentrations (**A**) and large cell ratio of platelet (**B**) affected by platelet lysate in different concentrations. Platelet lysate (**C**) and ristocetin (**D**) induced platelet aggregation in platelet-rich plasma. (**E**) Number of platelets per aggregate was induced by different concentrations of platelet lysate. In panels (**A**,**B**,**E**), ristocetin and 50% saline were used as positive and negative controls, respectively, and *n* = 10 in each experimental group. PC-PL, platelet concentrate–platelet lysate. PC-PL 2.5–10% indicates different platelet lysate fractions in the mixture. * *p* < 0.05 versus the saline group in the Student’s *t*-tests.

**Figure 2 jcm-11-03972-f002:**
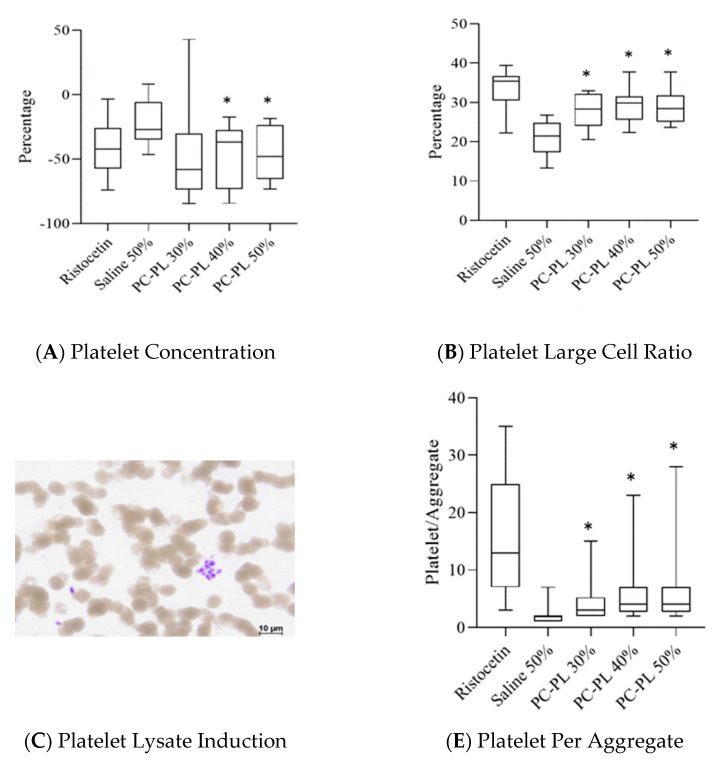
Effects of platelet lysate on platelet aggregation in direct mixing of platelet lysate with whole blood. Percentage changes in platelet concentrations (**A**) and large cell ratio of platelet (**B**) affected by platelet lysate of different concentrations. Platelet lysate- (**C**) and ristocetin- (**D**) induced platelet aggregation in platelet-rich plasma. (**E**) Number of platelets per aggregate induced by different concentrations of platelet lysate. Controls, sample size, statistical analysis and abbreviations used are as described in the legend to Figure 1. * *p* < 0.05 versus the saline 50% group in the Student’s *t*-tests.

**Figure 3 jcm-11-03972-f003:**
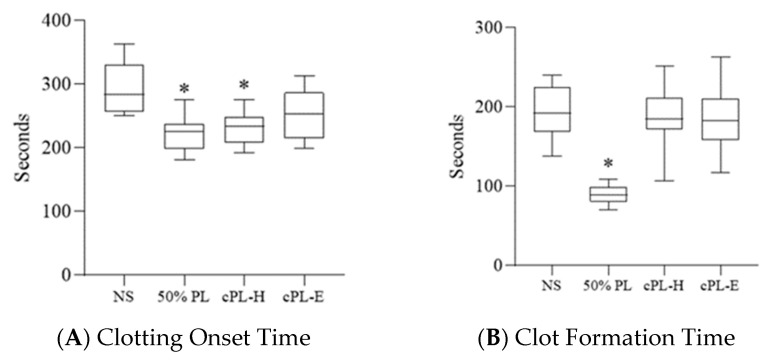
Effects of platelet lysate and fibrinogen-depleted platelet lysate on blood clot formation measured by rotational thromboelastometry. (**A**) Clotting onset time. (**B**) Clot formation time. (**C**) α angle. (**D**) Maximum clot firmness. In all experiments, *n* = 10 in each experimental group. * NS, normal saline; PL, platelet lysate; cPL-H, commercial platelet lysate-HELIOS; cPL-E, commercial platelet lysate-Evo. *p* < 0.001 versus the NS group in the Student’s *t*-tests.

**Table 1 jcm-11-03972-t001:** Changes in platelet concentration, platelet large cell ratio, and platelet per aggregate in different constituents of platelet lysate and platelet-rich plasma.

	Ristocetin	10% Saline	Platelet Lysate		
			2.50%	5%	10%
Percentage change of platelet concentration	−83.1 ± 27%	−1.5 ± 3%	−16.3 ± 13% *	−24.3 ± 22% *	−39.8 ± 26% *
Platelet large cell ratio	32.6 ± 5%	20.5 ± 3%	23.1 ± 3%	25.0 ± 5% *	32.1 ± 4% *
Platelet per aggregate	3.9 ± 2.7	2.4 ± 0.9	2.8 ± 1.8 *	3.5 ± 2.3 *	4.3 ± 3.2 *

Data are summarized from Figure 1. * *p* < 0.05 compared with the saline group.

**Table 2 jcm-11-03972-t002:** Changes in platelet concentration, platelet large cell ratio, and platelet per aggregate in different constituents of platelet lysate and whole blood.

	Ristocetin	50% Saline	Platelet Lysate		
			30%	40%	50%
Percentage change of platelet concentration	−41.1 ± 22%	−21.8 ± 19%	−46.1 ± 38%	−45.3 ± 24% *	47.1 ± 20% *
Platelet large cell ratio	33.4 ± 5%	20.8 ± 4%	28.0 ± 4% *	29.2 ± 5% *	29.0 ± 5% *
Platelet per aggregate	15.7 ± 10.0	1.7 ± 0.9	6.0 ± 5.2 *	5.3 ± 3.9 *	4.5 ± 3.1 *

Data are summarized from Figure 2. * *p* < 0.05 compared with the saline group.

## Data Availability

Not applicable.

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
