# Peer review of "Risks in Induction of Platelet Aggregation and Enhanced Blood Clot Formation in Platelet Lysate Therapy: A Pilot Study"

_jcm, 2022, doi:10.3390/jcm11143972_

Round 1
Reviewer 1 Report
Using mostly in vitro approaches, Wen et al. assessed the thrombotic risk associated to therapies that utilize platelet lysates (PL). Briefly, the authors mixed PLs derived from platelet concentrates (PCs) with platelet-rich plasma (PRP) or whole blood, and then proceeded to measure platelet activation and aggregation, hypercoagulability, clot time formation and clot firmness, among other parameters. The results showed that PLs induced platelet aggregation and cause a lower clotting onset and clot formation times, and greater clot firmness.
Although the data need to be complemented by in vivo models in the future, the in vitro results already reveal some of the obvious risks and side effects of injecting PLs and PCs.
Author Response
Using mostly in vitro approaches, Wen et al. assessed the thrombotic risk associated to therapies that utilize platelet lysates (PL). Briefly, the authors mixed PLs derived from platelet concentrates (PCs) with platelet-rich plasma (PRP) or whole blood, and then proceeded to measure platelet activation and aggregation, hypercoagulability, clot time formation and clot firmness, among other parameters. The results showed that PLs induced platelet aggregation and cause a lower clotting onset and clot formation times, and greater clot firmness.
Although the data need to be complemented by in vivo models in the future, the in vitro results already reveal some of the obvious risks and side effects of injecting PLs and PCs.
Authors’ reply: We thank the reviewer for the favorable comment.
Reviewer 2 Report
This is a very interesting study that demonstrates the platelet lysate-induced hypercoagulability, highlighting the risk of thrombosis in some clinical situations.
The manuscript is well written, however there are some points that needs to be improved.
In tha Abstract, line 26, please define which are the controls.
In Line 65: please replace the word "however" with "On the other hand"
In Line 78: Why healthy volunteers are twenty but the samples studied are ten?
In Line 130: Please delete "in this work"
In Line 134: Please replace the dot with a comma
In Line 142: Please delete the "s" in the word "Figures"
In Line 150: Please add the word "respectively" after the word "controls"
In title of Table 1 and 2, please remove the "Summary of"
In Line 205: please "D"
In Line 227: please replace the word "this" with "the"
In Line 236: please replace "undergoing" with the word "undergo"
In Line 237: please delete the word "in"
In Line 247: please delete the word "is"
In Tables 1 and 2, did data represent the average? which is the range of the samples? please add.
In statistical analysis, it would be better to use a statistical program (such as SPSS) instead of Microsoft excel in order to have more accuracy in data processing.
Because of the small sample number, I would suggest to be referred somehow in the title of the article, that this is a pilot study.
Author Response
- This is a very interesting study that demonstrates the platelet lysate-induced hypercoagulability, highlighting the risk of thrombosis in some clinical situations.
Author’s reply: Thank you.
- The manuscript is well written, however there are some points that needs to be improved.
In the Abstract, line 26, please define which are the controls.
Author’s reply: We have revised the statement as: “Compared with the controls, which were 1:1 mixtures of normal saline and whole blood, ... ”.
- In Line 78: Why healthy volunteers are twenty but the samples studied are ten?
Author’s reply: We have revised the statement as: “Ten samples were used in platelet aggregation assays, and the other in ROTEM assays.”
- In Tables 1 and 2, did data represent the average? which is the range of the samples? please add.
Author’s reply: The data represent the average values. We have now added the range of the samples in Tables 1 and 2.
- In statistical analysis, it would be better to use a statistical program (such as SPSS) instead of Microsoft excel in order to have more accuracy in data processing.
Author’s reply: We have repeated the statistical analysis using SPSS, and have revised the statement in section 2.4. Statistical Analysis as: “Statistical analysis was performed using SPSS (version 20; SPSS Inc.).”.
- Because of the small sample number, I would suggest to be referred somehow in the title of the article, that this is a pilot study.
Author’s reply: A good suggestion. The revised title now reads: “Risks in Induction of Platelet Aggregation and Enhanced Blood Clot Formation in Platelet Lysate Therapy: a Pilot Study.”
- In Line 65: please replace the word "however" with "On the other hand"
- In Line 130: Please delete "in this work"
- In Line 134: Please replace the dot with a comma
- In Line 142: Please delete the "s" in the word "Figures"
- In Line 150: Please add the word "respectively" after the word "controls"
- In title of Table 1 and 2, please remove the "Summary of"
- In Line 205: please "D"
- In Line 227: please replace the word "this" with "the"
- In Line 236: please replace "undergoing" with the word "undergo"
- In Line 237: please delete the word "in"
- In Line 247: please delete the word "is"
Author’s reply: All the above comments have been revised as suggested.
Reviewer 3 Report
Dear Authors, the manuscript entitled " Risks in Induction of Platelet Aggregation and Enhanced Blood 2 Clot Formation in Platelet Lysate Therapy" represents a valuable study to the field. However, major revisions are required before processed to the next step. Below you can find my comments
1) In the introduction section, lines 35-37. The first sentence must be totally removed, the classification of PCs is not accurate enough.
2) Introduction, the authors should describe better the aggregation potential and thrombus formation, which is related with the use of PL and PC. Howver, and based to our experience, it is difficult to occur thrombosis after the administration of PL. I don't think that there is huge possibility to be performed this adverse event.
3) Introduction,lines 69-74, the aim of this study may confuse the readers. Please, describe better the method that you will follow to assess the aggregation potential of PRP and PL, to introduce the readers. Then describe the aim of this study appropriately.
4) In materials and methods, the authors should validate the quality characteristics of PL and PC, including, pH, lactate, fibrinogen, APT and PT in order to validate their thrombotic potential. Maybe PCS and PLs with low APTT, PT and fibrinogen levels cannot cause the thrombus.
5) In materials and methods, the authors must quantify the growth factor levels in PC and PL using commercial ELISA kits.
5) In Figure 1 C and D and also in Figure 2 C and D, there are no increased number of aggregations.
6) In the discussion section the authors, should include more studies and to compare their results.
Author Response
Dear Authors, the manuscript entitled " Risks in Induction of Platelet Aggregation and Enhanced Blood 2 Clot Formation in Platelet Lysate Therapy" represents a valuable study to the field. However, major revisions are required before processed to the next step. Below you can find my comments
- 1) In the introduction section, lines 35-37. The first sentence must be totally removed, the classification of PCs is not accurate enough.
Author’s reply: The first sentence of the introduction section was intended to address the relationship between “PC” and “PRP” with publication cited. Hence, we think it is justified for us to keep the sentence.
- 2) Introduction, the authors should describe better the aggregation potential and thrombus formation, which is related with the use of PL and PC. However, and based to our experience, it is difficult to occur thrombosis after the administration of PL. I don't think that there is huge possibility to be performed this adverse event.
Author’s reply: PL and PC contain cytokines known to be involved in platelet aggregation and thrombus formation, and there is only irreversible blindness caused by ophthalmic artery occlusion reported in patients after PC injection into facial skin. This observation has led us to draw a parallel and make the assumption that PL would possibly cause thrombosis. Although our results revealed potential PL-induced thrombosis, we do mention at the end of the Discussion section that animal studies are needed to further substantiate our proposition.
In the revision, we have replaced the sentence “However, evidence is lacking to support potential thrombosis in these cases.” by “This observation has led us to draw a parallel and make the assumption that PL would possibly cause thrombosis.” in lines 68-69.
- 3) Introduction, lines 69-74, the aim of this study may confuse the readers. Please, describe better the method that you will follow to assess the aggregation potential of PRP and PL, to introduce the readers. Then describe the aim of this study appropriately.
Author’s reply: Hemostasis involves three basic steps: vascular spasm, the formation of a platelet plug (platelet aggregation), and coagulation (blood clot formation). To approach whether PL causes thrombosis, we divided the study into three parts:
First part: To determine whether PL induces platelet aggregation through direct mixing of PL with PRP and PL with the whole blood (lines 69-71). Second part: To investigate whether PL induces blood clot formation via rotational thromboelastometry (lines 71-72). Third part: Whether thrombotic risk could be reduced by use of fibrinogen depletion and anticoagulants (lines 72-74).
In the revision, we have added the sentence “To investigate whether PL causes thrombosis, we divided the study into three parts—platelet aggregation, blood clot formation and prevention of hypercoagulation.” in lines 73-74.
- 4) In materials and methods, the authors should validate the quality characteristics of PL and PC, including, pH, lactate, fibrinogen, APT and PT in order to validate their thrombotic potential. Maybe PCs and PLs with low APTT, PT and fibrinogen levels cannot cause the thrombus.
Author’s reply: Because PCs were obtained from Taiwan Blood Services Foundation (a non-remunerated blood donation system and sole supplier of medical therapeutic fresh blood products in Taiwan), we validated the basic quality of these PC supplies through platelet concentration 〔>2.75 × 1010 per bag (250 ml)〕. The PLs were validated by platelet aggregometry test (by light transmission aggregometry; data not shown). However, PLs cause platelet aggregation too fast to be measured by light transmission aggregometry. Fortunately, and thanking you for reminding this critical point of the study, we found that PLs also caused a decrease in platelet concentration in platelet aggregometry test to indicate and validate the thrombotic potential of PLs.
The statements in lines 85-91 were revised as follows:
PL was produced from shelf-life-expired PCs obtained from the Taiwan Blood Services Foundation, which is a non-remunerated blood donation system. The basic quality of platelet concentrations >2.75 × 1010 per bag (250 ml) of the PC supplies was assured by a hematological analyzer XN-9000 (Sysmex Corp.; Japan) (data not shown). On the other hand, PLs induce platelet aggregation too fast to be measured by light transmission aggregometry. Nonetheless, a decrease in platelet concentration was consistently found in PLs in the test to support the thrombotic potential of PLs.
- 5) In materials and methods, the authors must quantify the growth factor levels in PC and PL using commercial ELISA kits.
Author’s reply: Because growth factors play a minor role in the various steps of platelet aggregation or blood clot formation, we have no data on the growth factor levels in PC and PL.
- 6) In Figure 1 C and D and also in Figure 2 C and D, there are no increased number of aggregations.
Author’s reply: In Figure 1 C and D and also in Figure 2 C and D, we intended to demonstrate increased number of platelets per aggregate induced by PL and ristocetin (positive control).
- 7) In the discussion section the authors, should include more studies and to compare their results.
Author’s reply: In the revised title, we have added “a pilot study” to indicate that this is a first study to investigate and address the thrombotic risk of PL therapy.
Round 2
Reviewer 3 Report
Dear Authors,
Thank you for providing the revised version of your manuscript. However, a Number of my comments have not been well answered by your side, hence require further clarification.
1) In the introduction section, lines 35-37. The first sentence must be totally removed, the classification of PCs is not accurate enough. In my opinion this is not an accurate classification of PCs and this sentence may provide false information to the readers and the scientific Society. Please remove or modify this sentence to be more accurate.
2) In materials and methods, the authors should validate the quality characteristics of PL and PC, including, pH, lactate, fibrinogen, APT and PT in order to validate their thrombotic potential. Maybe PCs and PLs with low APTT, PT and fibrinogen levels cannot cause the thrombus.
The authors have not successfully addressed this comment. Please it is very important for the readers to provide information regarding the quality characteristics of the produce PL and PC used in this study. Thee are analysers such as the gas analysers that can detect and quantitatively measure the above quality characteristics both in PL and PC.
3)In materials and methods, the authors must quantify the growth factor levels in PC and PL using commercial ELISA kits.
The authors also did not perform assays to quantify the growth factor content of the produce PL and PC. The manuscript has been submitted to a highly impact journal, so to perform advanced approaches in order to provide significant evidence to the readers is very crucial. The authors should provide evidence regarding the growth factor content of PLs and PCs used in this study.
In the discussion section the authors should include more studies to compare their results in order to discuss better their data and to Support their results.
Overall, the authors should address the above comments and provide answers step by step in order the manuscript to improved in quality and proceed to the next step of the publication process.
Author Response
Dear Authors,
Thank you for providing the revised version of your manuscript. However, a Number of my comments have not been well answered by your side, hence require further clarification.
1) In the introduction section, lines 35-37. The first sentence must be totally removed, the classification of PCs is not accurate enough. In my opinion this is not an accurate classification of PCs and this sentence may provide false information to the readers and the scientific Society. Please remove or modify this sentence to be more accurate.
Author’s reply: Sorry, we are not intended to provide false information to the readers and the scientific society. Please provide accurate reference to let us modify those sentence, thank you!
2) In materials and methods, the authors should validate the quality characteristics of PL and PC, including, pH, lactate, fibrinogen, APT and PT in order to validate their thrombotic potential. Maybe PCs and PLs with low APTT, PT and fibrinogen levels cannot cause the thrombus.
The authors have not successfully addressed this comment. Please it is very important for the readers to provide information regarding the quality characteristics of the produce PL and PC used in this study. There are analyzers such as the gas analyzers that can detect and quantitatively measure the above quality characteristics both in PL and PC.
Author’s reply: Because these characteristics of PL and PC (pH, lactate, fibrinogen, APT and PT) must be measured in fresh sample, we found it impossible to further address the comments.
3)In materials and methods, the authors must quantify the growth factor levels in PC and PL using commercial ELISA kits.
The authors also did not perform assays to quantify the growth factor content of the produce PL and PC. The manuscript has been submitted to a highly impact journal, so to perform advanced approaches in order to provide significant evidence to the readers is very crucial. The authors should provide evidence regarding the growth factor content of PLs and PCs used in this study.
Author’s reply: Because we did not quantify the growth factor levels in PC and PL when performing the experiments of this study, the levels of growth factors will be distorted when using the frozen-stored samples to perform assays.
In the discussion section the authors should include more studies to compare their results in order to discuss better their data and to Support their results.
Author’s reply: Because this is the first study to investigate and address the thrombotic risk of PL therapy, we found no similar study to be compared. And we have added “a pilot study” to the revised title.